# Interplay between oceanic subduction and continental collision in building continental crust

Di-Cheng Zhu[1] ✉, Qing Wang[1], Roberto F. Weinberg [2], Peter A. Cawood [2], Sun-Lin Chung [3], Yong-Fei Zheng[4], Zhidan Zhao[1], Zeng-Qian Hou[5] & Xuan-Xue Mo[1]

Generation of continental crust in collision zones reflect the interplay between oceanic subduction and continental collision. The Gangdese continental crust in southern Tibet developed during subduction of the Neo-Tethyan oceanic slab in the Mesozoic prior to reworking during the India-Asia collision in the Cenozoic. Here we show that continental arc magmatism started with fractional crystallization to form cumulates and associated medium-K calc-alkaline suites. This was followed by a period commencing at ~70 Ma dominated by remelting of pre-existing lower crust, producing more potassic compositions. The increased importance of remelting coincides with an acceleration in the convergence rate between India and Asia leading to higher basaltic flow into the Asian lithosphere, followed by convergence deceleration due to slab breakoff, enabling high heat flow and melting of the base of the arc. This two-stage process of accumulation and remelting leads to the chemical maturation of juvenile continental crust in collision zones, strengthening crustal stratification.

Continental collision zones generally develop from an initial phase of oceanic subduction generating continental arc magmatism to a phase of continental collision reworking the arc lithosphere[1–3]. These zones record the generation of continental crust of andesitic to dacitic bulk composition and record the key geodynamic processes leading to the growth and preservation of continental crust on Earth[4,5]. Continental arc magmatism before collision records recycling of the subducting oceanic crust, fluid-fluxed melting of the mantle wedge, and the generation of juvenile mafic crust[6]. In contrast, continental collision generally results in reworking of both juvenile and ancient continental crust, melting of the crust, and the formation of mature felsic crust[1,3]. However, it remains unclear how this reworking of continental arc lithosphere modifies the nature and composition of the crust and why this process occurs in collision zones.

The Gangdese magmatic belt in southern Tibet (Fig. 1a) was part of an accretionary orogen resulting from subduction of the Neo-Tethyan oceanic lithosphere that generated continental arc magmatism in the Mesozoic. It then became part of a collisional orogen as a result of the India-Asia collision in the Cenozoic[7–9]. The Gangdese belt is a superb site for studying the growth and reworking of continental crust along collision zones. This is because the entire magmatic record, from growth of the continental arc crust during the Mesozoic[8–10], to its reworking during the Cenozoic (Fig. 1b) is well-preserved, well-exposed, and well-dated[7–12]. Furthermore, the kinematic framework (i.e. the India-Asia convergence direction and rate as well as associated driving mechanisms) responsible for the formation of this belt is well-established[9,13,14]. The magmatic rocks in this belt can be subdivided into pre-collisional (>60 Ma), syn-collisional (60−45 Ma), and post-collisional (45−10 Ma) suites. This subdivision

[1]State Key Laboratory of Geological Processes and Mineral Resources, China University of Geosciences, 100083 Beijing, China. [2]School of Earth, Atmosphere and Environment, Monash University, Melbourne, VIC 3800, Australia. [3]Department of Geosciences, National Taiwan University, Taipei, China. [4]CAS Key Laboratory of Crust-Mantle Materials and Environments, School of Earth and Space Sciences, University of Science and Technology of China, Hefei 230026, China. [5]Institute of Geology, Chinese Academy of Geological Sciences, 100037 Beijing, China. ✉e-mail: dchengzhu@163.com

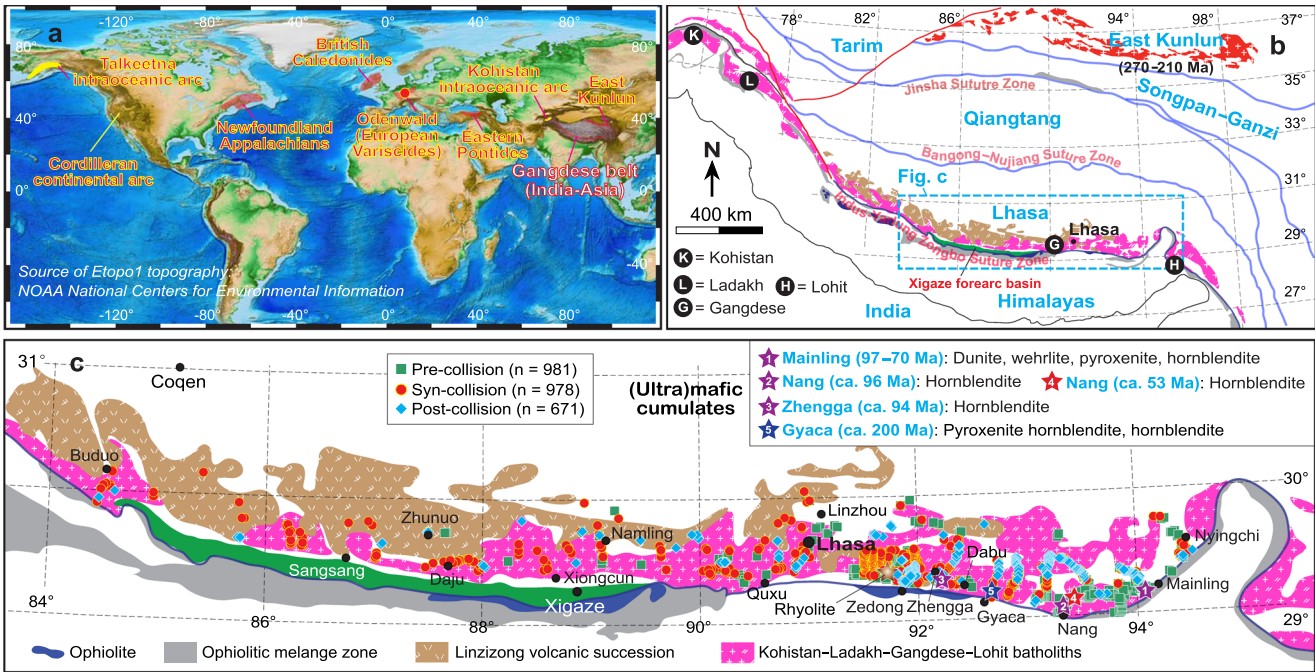

**Fig. 1 | Present-day topography of the Earth and the tectonic framework of the Tibetan Plateau. a** Etopo1 topography showing the locations of major collisional zones and magmatic arcs. **b** Location of the Gangdese magmatic belt (including the Gangdese Batholith and Linzizong volcanic succession) in the context of the Tibetan Plateau[7]. **c** Sample localities of the Gangdese Batholith in this study[8] as well as the outcrops of (ultra)mafic cumulates marked by colored and numbered star symbols.

is based on the multiple lines of evidence that constrain the timing of initial collision between India and Asia to ~60 Ma[15,16]. It is also based on post-collision being defined as the time following complete loss of the pull force from the subducting oceanic slab due to its breakoff, which terminated at ~45 Ma[8].

In order to explore the processes of generation and reworking of continental crust in continental collision zones, we focus on the 100−10 Ma old magmatic rocks from the central to eastern Gangdese belt (longitude E84°−E95°) (Fig. 1c). Our dataset consists of 2630 plutonic samples including new and published data of whole-rock and mineral geochemistry, zircon U-Pb age and Hf-O isotopes (Supplementary Table 1−8). The data contain 981 pre-collisional, 978 syn-collisional, and 671 post-collisional plutonic samples with an excellent spatial coverage along and across the strike of the belt, effectively reducing potential sample bias in the estimation of average crustal composition.

## Results and discussion
### Pre-collisional suites dominated by fractional crystallization of mantle-derived magmas
The pre-collisional magmatic rocks form batholiths dominated by hornblende-rich rocks with coeval ultramafic cumulates[9,17,18]. The temporal changes in the nature of magmatism are shown by whole-rock composition plots using bivariate kernel density (Fig. 2a−c). The pre-collisional suite defines a Z-shaped trend in the plot of Mg# vs $SiO_2$ (Fig. 2a) and an S-shaped trend in the plot of $Al_2O_3$ vs $SiO_2$ (Fig. 2b). These trends are similar to those from the Kohistan and Talkeetna arcs[19,20], and result from the sequential accumulation of olivine (Ol) → orthopyroxene (Opx) + clinopyroxene (Cpx) → hornblende (Hbl) + Fe-Ti oxide → plagioclase (Pl)[21]. Such a sequence is controlled by hydrous fractional crystallization, resembling the Southern Plutonic Complex of the Kohistan arc[19] and the Chelan Complex of the incipient Cascades arc (Ol + Cpx → Cpx + Hbl → Hbl)[22], as well as results from $H_2O$-saturated crystallization experiments at 1.0 GPa[19]. This sequence differs from that of mature magmatic arcs, as exemplified by the Sierra Nevada in the North Cordillera, which

shows a sequence of crystallization from high-Mg pyroxenites (with minor Grt) to low-Mg, garnet- or plagioclase-rich pyroxenite depending on pressure[23]. These results indicate that the pre-collisional mafic rocks were derived from wet fractional crystallization of basaltic melts from the metasomatized mantle wedge[17,18] with high initial $H_2O$ content, likely in excess of 3.0 wt%[24,25] in an immature arc.

In such a hydrous system, hornblende forms initially through hydration of pyroxene, and continued crystallization of hornblende leads to significant enrichment of $SiO_2$ in the residual melt[20]. This mechanism may be responsible for the origin of pre-collisional medium-K, calc-alkaline, intermediate to felsic plutonic rocks in the Gangdese belt (Fig. 2c). This interpretation is supported by the existence of Late Cretaceous (~110−70 Ma) compositionally continuous cumulates (Supplementary Fig. 1) ranging from dunite to wehrlite, pyroxenite, hornblendite, hornblende gabbro, and gabbronorite (Fig. 1c) at the arc base, and the occurrence of granitoids at shallower levels[18,26].

### Syn-collisional suite dominated by remelting of arc crust
The syn-collisional suite is characterized by hornblende-poor rocks and range in composition from mafic to felsic. Whole-rock Mg# and $Al_2O_3$ versus $SiO_2$ plots for both syn- and post-collisional samples (Fig. 2d, e and g, h) lack the Z-shaped and S-shaped trends found for the pre-collisional samples. This is because of the absence of dunitic to pyroxenitic rocks among these samples. Other striking features of the syn-collisional samples include: (1) small amounts of hornblendites and cumulate hornblende gabbros suggesting limited accumulation, (2) widespread 58−45 Ma mafic microgranular enclaves and dykes indicative of basaltic injections and magma mixing/mingling[8–10], (3) voluminous high-K felsic rocks ($SiO_2 > 60$ wt%) (Fig. 2f), and (4) cumulate hornblende gabbro (~50 Ma) with petrographic relationships indicating crystallization starting with olivine and ending with hornblende along this sequence: Ol → Opx → Opx + Pl → Hbl + Fe-Ti oxide (Fig. 2j, k). The appearance of plagioclase before hornblende is typical of damp fractional crystallization[19,24] of mantle-derived melts

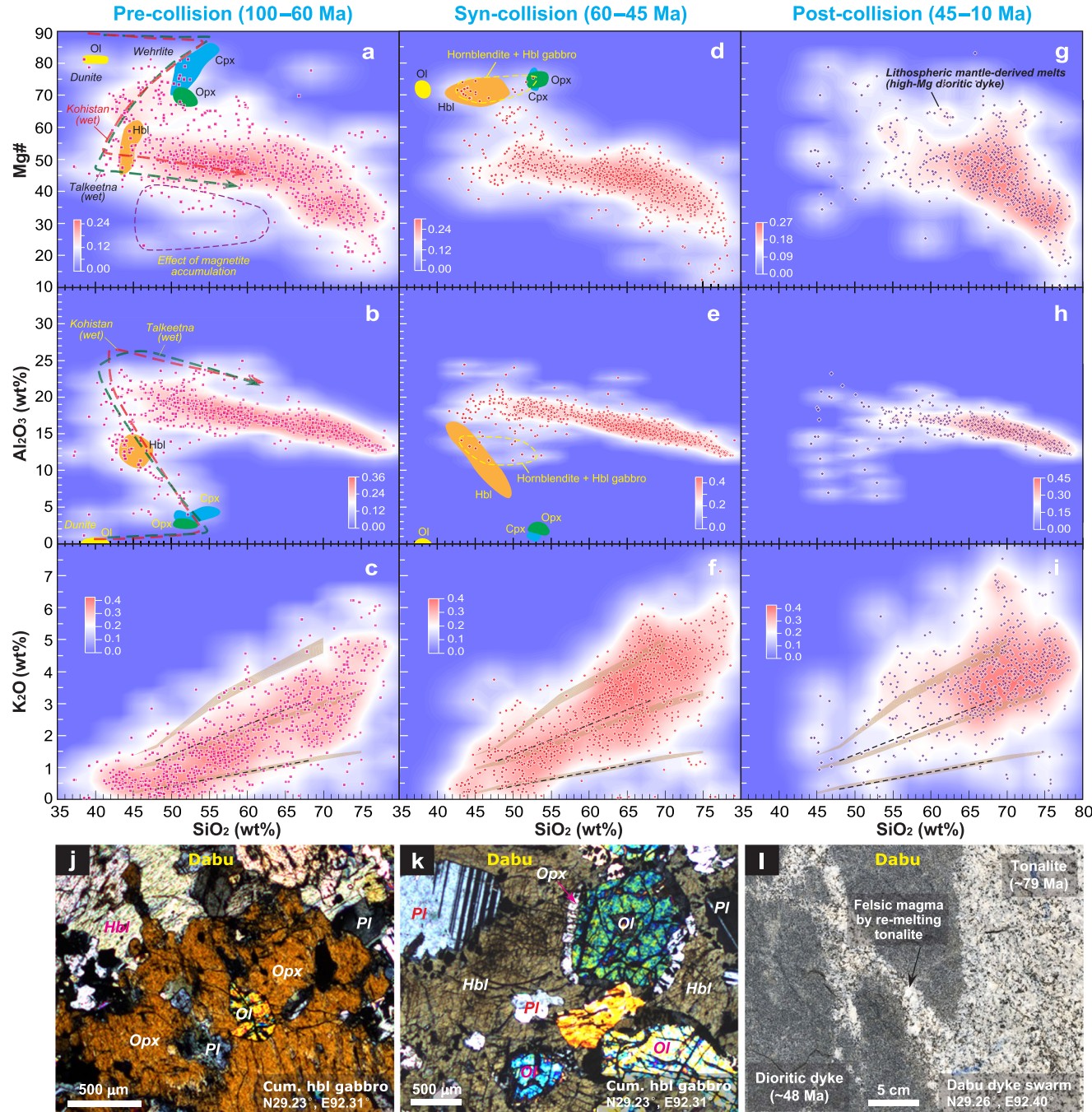

**Fig. 2 | Whole-rock Mg#, Al₂O₃, and K₂O vs. SiO₂ plots and photomicrographs of samples from the Gangdese Batholith. a–c** Plots of pre-collisional samples (100–60 Ma). Mafic and ultramafic samples define a Z-shaped trend in **a** and an S-shaped trend in **b**, similar to Kohistan (dashed red line) and Talkeetna (dashed green line) trends[19,20]. **d–f** Plots of syn-collisional samples (60–45 Ma). **g–i** Plots of post-collisional samples (45–10 Ma). Colored background indicates sample density distribution as measured by bivariate kernel density, where red background corresponds to increased sample concentration or density. The same is true for Fig. 3. Comparison between **c**, **f**, and **i** shows the temporal evolution of the magmatic rocks towards higher silica and higher potassium contents, accompanied by the loss of samples with SiO₂ < 55 wt%. Photomicrographs of a syn-collisional, ~50 Ma

Hbl gabbro from Dabu, where **j** shows Pl included in Opx surrounded by Hbl, and **k** shows Hbl formed by Opx + liquid reaction and Opx rims around Ol caused by the peritectic reaction Ol + liquid = Opx. These relationships indicate a damp environment where Hbl is last to crystallize at the end of the sequence Ol → Opx → Opx + Pl → Hbl. **l** Detail at the contact between a syn-collisional dioritic dyke (~48 Ma) and an older tonalite (~79 Ma) resulting in the remelting of the tonalite and back-veining of the dyke. Mg# = molar 100×Mg²⁺/(Mg²⁺+TFe²⁺), where TFe²⁺ represents total Fe. Whole-rock and mineral compositions are given in Supplementary Table 1, and 3. Mineral abbreviation: Ol olivine, Cpx clinopyroxene, Opx orthopyroxene, Hbl hornblende, Pl plagioclase.

with low initial H₂O contents of 1–2 wt%[27] compared to >3 wt% H₂O for wet fractional crystallization[24,25].

In a damp system, significant SiO₂ enrichment is suppressed by the peritectic reaction Ol + melt = Opx[20] (Fig. 2k). This means that the voluminous syn-collisional, high-K calc-alkaline, intermediate to felsic

plutonic rocks in the Gangdese belt cannot be ascribed to fractional crystallization. The Linzizong calc-alkaline, high-Al₂O₃ basalts and K-rich basalts (55–50 Ma)[28], as well as the Gangdese syn-collisional mafic plutons (58–45 Ma) (Fig. 2d, e), suggest repetitive injection of basaltic magmas during continental collision, causing melting of the

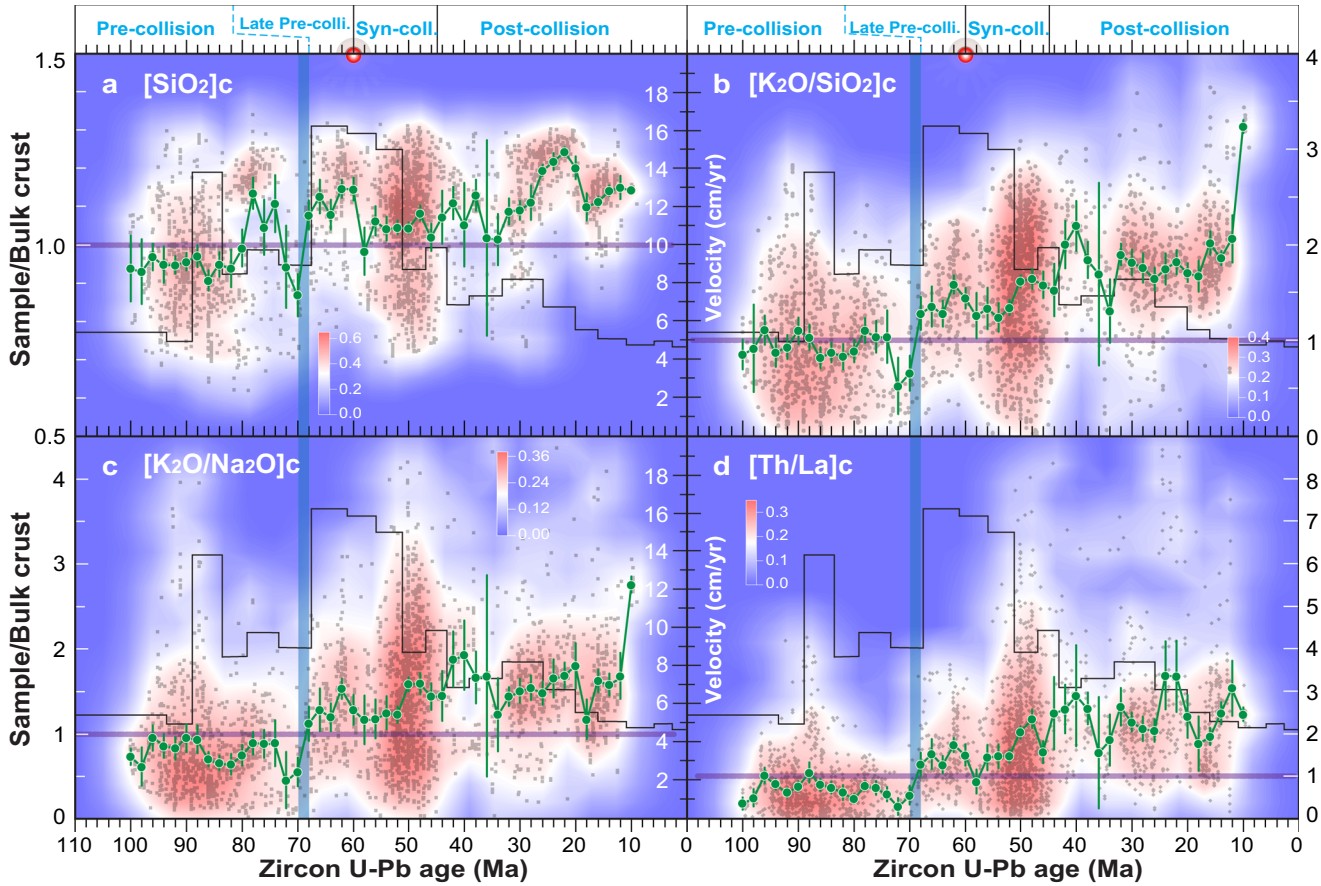

**Fig. 3 | Evolving composition of magmatic rocks from the Gangdese Batholith.**
**a–d** Gray dots show composition of individual samples and larger green dots with green vertical lines are average compositions and 2 s.e. uncertainties of samples in 2 Myr bins (obtained using the Monte Carlo analysis with weighted bootstrap resampling)[37] normalized by bulk continental crust. Values of bulk[38] are shown by purple horizontal lines. The India-Asia convergence velocity since 110 Ma (black line) is from ref. 13 and is shown for comparison with the changes in magmatic compositions. Notice an abrupt increase in the India-Asia convergence velocity at ~70 Ma, contemporaneous with the change in chemistry of the magmatic rocks, and post-dating a period of magma paucity[14]. This acceleration is inferred to mark the onset of slab rollback[8,9,14]. Notice also the concentration of data at ~50 Ma suggesting a magmatic flare-up, coeval with the slowdown of the Indian plate, both of which are inferred to be related to slab breakoff [8].

pre-existing crust. As these mafic magmas crystallize Ol + Px, $H_2O$ concentrates in the melt but is prevented from increasing significantly because it diffuses out into the surroundings during slow crystallization in a hot crust[29–31]. This would cause the magma to remain damp (1–2 wt%[27]), rather than wet (>3 wt% $H_2O$[24,25]), and also cause water-fluxed melting of the pre-existing crust to generate felsic magmas, that would mix with the newly-injected mafic magmas[30–33]. Field evidence for melting is shown by back-veining of a ~48 Ma dioritic dyke by felsic magma derived from the melting of its ~79 Ma wall-rock (Fig. 2l), and hybridization is indicated by widespread mafic microgranular enclaves containing entrained coarse-grained K-feldspar and quartz xenocrysts from the host felsic magma[8,9,34].

### Post-collisional suite dominated by remelting of arc crust plus subducted Indian continental material
Partial melting of pre-existing intrusions and mixing, accompanied by some fractional crystallization may also be applicable to the post-collisional suite, dominated by high-K felsic rocks ($SiO_2$ > 60 wt%) (Fig. 2g–i). This is because: (1) the formation of volumetrically insignificant post-collisional hornblende gabbros may indicate a diminished role of fractional crystallization, (2) the early appearance of plagioclase followed by hornblende in hornblende gabbro[35] also suggests damp fractionation, (3) evidence for Miocene migmatization of syn-collisional dioritic gneisses accompanied by post-collisional granitic rocks with xenocrysts from the dioritic gneisses, collectively recording anatexis[36], and (4) more enriched radiogenic isotope compositions compared to the pre- and syn-collisional samples, along with high Cr and Ni contents, and Mg# (Fig. 2g) in some Miocene intermediate-to-felsic samples, indicate hybridization between melts derived from the pre-existing Gangdese crust and ultrapotassic magmas from the Gangdese lithospheric mantle metasomatized by subducted Indian continental material[12].

### Compositional changes from subduction to collision
In order to explore temporal changes in the average chemical composition of magmatic rocks from the Gangdese belt, we employ Monte Carlo analysis with weighted bootstrap resampling approach. This approach effectively minimizes sampling bias and achieves a best estimate of the average composition of exposed continental crust through time[37]. The results, coupled with the bivariate kernel density of sample distribution, are illustrated in Fig. 3. The most intriguing finding is a compositional shift at ~70 Ma from values below to values above the estimated composition of bulk continental crust[38] in terms of $SiO_2$, $K_2O/SiO_2$, $K_2O/Na_2O$, and Th/La ratios (Fig. 3). This shift is not reflected concurrently by zircon Hf-O isotope values. Zircon $\varepsilon_{Hf}(t)$ displays a delayed decrease in values that occurs only at ~55 Ma (Fig. 4a), while zircon $\delta^{18}O$ values remain nearly constant ranging between 5.5–7.2‰ from 100 to 30 Ma (Fig. 4b), typically ~1‰ higher than igneous zircons in equilibrium with mantle-derived magmas (5.3 ± 0.6‰, 2σ[39]). The compositional shift in whole-rock geochemistry

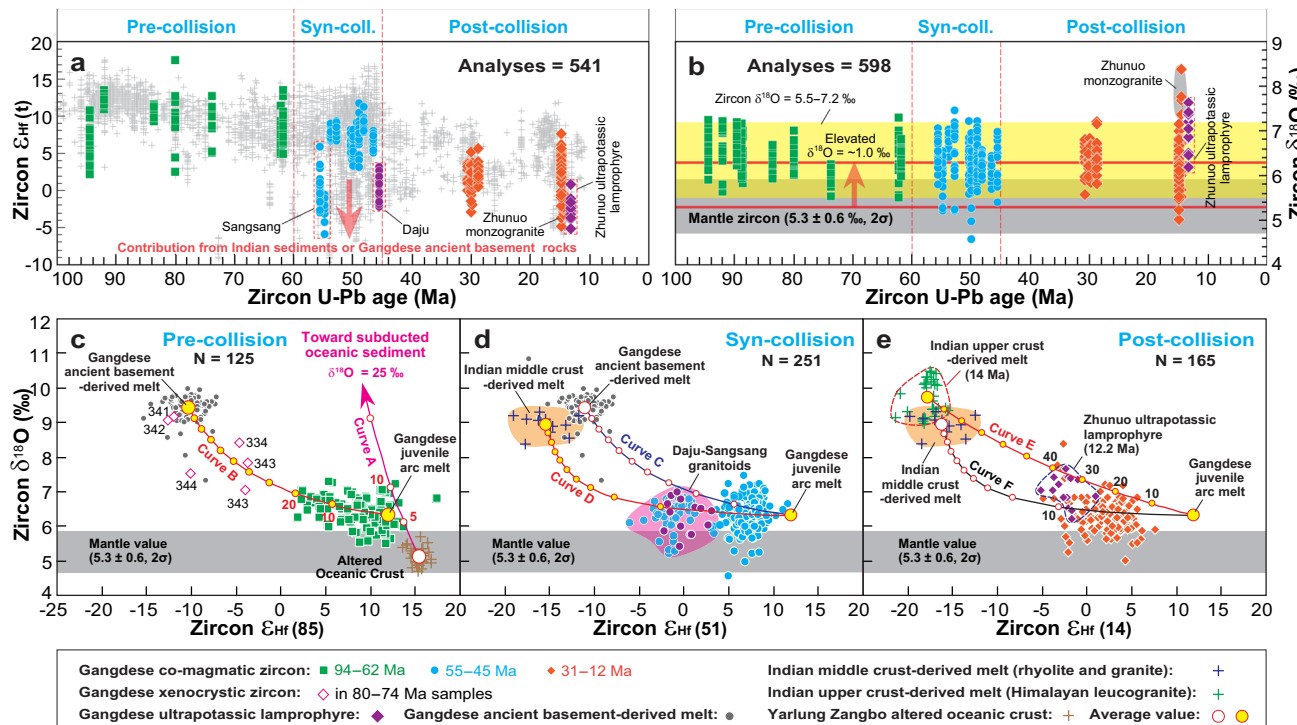

**Fig. 4 | Evolving zircon Hf-O isotope compositions of magmatic rocks from the Gangdese Batholith. a** Temporal change in zircon $\varepsilon_{Hf}(t)$ showing that some syn- and post-collisional samples have lower, more enriched values. **b** Temporal change in zircon $\delta^{18}O$ showing a limited range of 5.5–7.2 ‰ (yellow band) with a number of exceptions with values above 7.2 ‰, particularly among Miocene post-collisional samples. The horizontal thick red lines indicate the ~1‰ systematic increase in zircon $\delta^{18}O$ of the 100–30 Ma samples relative to mantle values (5.3 ± 0.6‰, 2σ[39]), suggesting ~5% subducted sediment addition (Curve A in **c**). **c**–**e** $\delta^{18}O$ versus $\varepsilon_{Hf}(t)$ for zircons from the Gangdese samples. Curves B and C of binary mixing in **c** and **d** linking the Gangdese juvenile arc melt with the Gangdese ancient basement-derived melt suggest crustal assimilation. Curves A of binary mixing in **c** linking the altered oceanic crust with the subducted oceanic sediment indicate recycling of subducted sediment, and curves D, E, and F of binary mixing in **d** and **e** linking the Gangdese juvenile arc melt with the Indian middle or upper crust-derived melts denote recycling of Indian supracrustal material. The entire zircon $\varepsilon_{Hf}(t)$ dataset from the Gangdese Batholith (gray crosses) is plotted in **a** as background. All Hf isotope ratios were age-corrected to 85 Ma in **c**, 51 Ma in **d**, and 14 Ma in **e**. Small circles on the curves represent 10% AFC increments except for the one circle labeled 5% close to the Gangdese juvenile arc melt. Number beside red diamonds (e.g., 343) in **c** represents the age of the xenocrystic zircon. All available data of zircon Hf-O isotopes in **a** and **b** are measured at the same spots or the same domains used for age determinations of concordant individual analyses (Supplementary Table 4). The methods for selecting Hf abundances and Hf-O isotopic compositions of each end-member are given in Supplementary Materials.

may represent a shift in the average exposure level of the arc crust to shallower crust since ~70 Ma[40]. However, we consider it more likely that this shift reflects changes in magma source, magmatic processes, and geodynamics, as discussed in the following sections.

## Role of recycled supracrustal component in generating the compositional change

The isotopic changes of arc magmas have generally been interpreted to result from increased input of subducted sediments, as proposed for syn-collisional rocks in southern Sulawesi (Indonesia)[41], and/or enhanced involvement of ancient basement rocks from the overriding plate, as suggested for arc rocks from the Cordilleran orogenic systems[42]. For the Gangdese Batholith, the fact that the zircon $\delta^{18}O$ of pre-collisional samples is overall higher than zircons in equilibrium with the mantle (median at 6.4 ± 0.2‰, 182 analyses, compared to 5.3 ± 0.6‰, 2σ[39]) suggest the addition of ~5 wt% of subducted sediment (Curve A in Fig. 4c; see Supplementary Materials for modeling assumptions). In parallel, the data also show a gentle increase in $\delta^{18}O$ accompanied by a decrease zircon $\varepsilon_{Hf}(t)$ that cannot be explained by subduction sediments alone and requires 0–20 wt% from Gangdese ancient basement (compare Curves A and B in Fig. 4c). This interpretation is further supported by the presence of ~340 Ma xenocrystic zircons in some 80–74 Ma samples (red diamonds in Fig. 4c).

For syn-collisional samples, the nearly unchanged zircon $\delta^{18}O$ is accompanied by decreasing zircon $\varepsilon_{Hf}(t)$ (Fig. 4d). This results from 0–10 wt% input of Indian crust, rather than the involvement of the

Gangdese ancient basement materials (compare Curves C and D in Fig. 4d). While assimilation of Indian material can explain the isotopic composition of syn-collisional magmatism (Fig. 4d), it is unlikely to account for the increase in $SiO_2$, $K_2O/SiO_2$, $K_2O/Na_2O$, and Th/La recorded by these samples. This is because the compositional changes occurred at 70–68 Ma, prior to the initial India-Asia collision (60–55 Ma[8,16]) and the onset of contribution from the Indian crust (~55 Ma; Fig. 4a). We ascribe this change in whole-rock composition to increased importance of remelting of arc rocks immediately preceding collision and continuing afterwards (next section).

Most post-collisional rocks have the same values of both zircon $\delta^{18}O$ and $\varepsilon_{Hf}(t)$ as earlier (Fig. 4e). Their origin and nature of the source can be explained as for the syn-collisional rocks. However, the Miocene lamprophyres form a separate group that have higher $\delta^{18}O$ and lower $\varepsilon_{Hf}(t)$. These rocks indicate the involvement of subducted Indian lithosphere (up to ~40 wt%; Curve E in Fig. 4e).

The median zircon $\delta^{18}O$ of 6.4 ± 0.2‰ (433 analyses) and calculated sediment contribution (~5 wt%) for the pre- and syn-collisional samples of the Gangdese belt are significantly lower than those for the Cretaceous gabbros and granitoids from the Sierra Nevada batholith with zircon $\delta^{18}O$ of 7.8 ± 0.7‰, for which more than 18% supracrustal contribution has been inferred[43]. Such low supracrustal inputs in the Gangdese belt are inconsistent with crustal relamination, which involves the recycling of high-$\delta^{18}O$ buoyant subducted supracrustal rocks[44]. Thus, relamination can be discarded as the cause of changes in geochemistry at ~70 Ma (Fig. 3a–d).

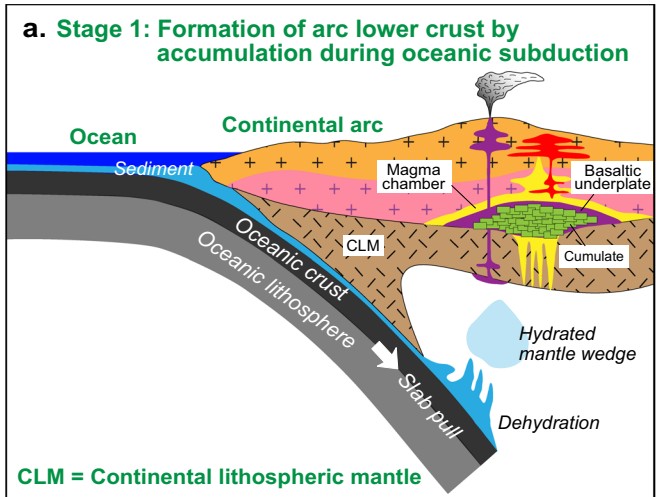

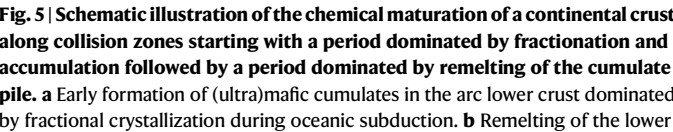

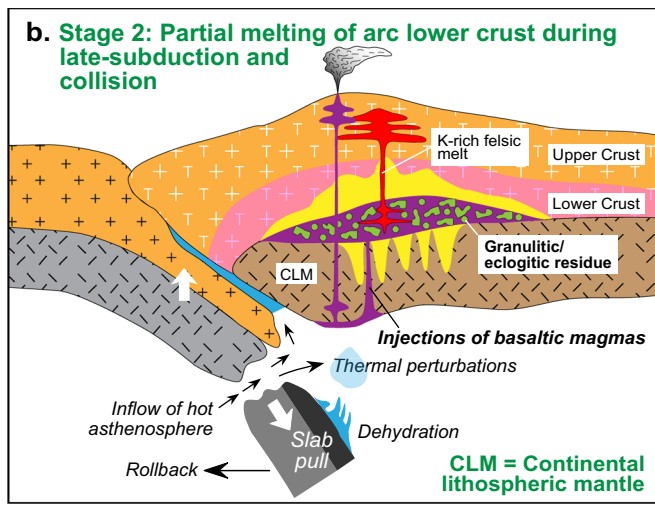

**Fig. 5 | Schematic illustration of the chemical maturation of a continental crust along collision zones starting with a period dominated by fractionation and accumulation followed by a period dominated by remelting of the cumulate pile. a** Early formation of (ultra)mafic cumulates in the arc lower crust dominated by fractional crystallization during oceanic subduction. **b** Remelting of the lower crust (including basaltic underplates and (ultra)mafic cumulates) to produce voluminous high-K, calc-alkaline and shoshonitic felsic rocks during late-subduction and collisional times. The CLM in **b** can also melt to produce shoshonitic melts. The yellow fields in **a** and **b** represent magma chamber and/or channel.

## Compositional changes caused by remelting pre-existing arc lower crust

Arc lower crustal rocks are composed mainly by underplated basaltic rocks and (ultra)mafic cumulates formed predominantly by fractional crystallization of mantle-derived magmas[17,18,45,46]. If enough potassium has been incorporated into such a mantle-derived magma due to melt metasomatism from subducted sediment, it may form a hornblende-rich cumulate (metamorphosed to amphibolite) when it stalls at the base of the crust[47]. Experimental studies and thermodynamic modeling show that partial melting of medium-to-high K basaltic/amphibolitic compositions can directly produce K-rich felsic melts at lower crustal conditions[33,47–49]. This is due to the nature of the melting reactions that tend to consume felsic minerals[47,50], as well as K-rich biotite and/or hornblende[33]. Similar to global continental arcs, basaltic underplates and cumulates from the Gangdese continental arc are characterized by high $K_2O$ contents (mostly > 0.5 wt.%) compared to the Kohistan-Ladakh oceanic arc[19,51] and to global oceanic arcs in general, typically having low $K_2O$ content (mostly <0.5 wt.%) (Supplementary Fig. 2). It follows that the remelting of the pre-collisional K-rich rocks in the Gangdese arc lower crust would have played a critical role in generating the compositional changes at ~70 Ma (Fig. 3) generating voluminous K-rich felsic magmas. Remelting would also account for the increase of Th/La in syn- and post-collisional magmatic rocks (Fig. 3d), as Th is more incompatible than La and concentrates in the melt during remelting[52].

Further evidence for remelting of arc rocks since 70 Ma is found in the following igneous rocks: (1) the Late Cretaceous (95–85 Ma) Mainling metagabbros with ~68 Ma irregular leucosomes generated at pressure of >1.0 GPa[53]; (2) the ~52 Ma Zedong rhyolite (Fig. 1c) with ~63 Ma inherited cores (Supplementary Fig. 3), indicating the melting of ~63 Ma rocks at ~52 Ma, similar to the ~50 Ma melting of the ~58 Ma rocks recorded by the Ladakh batholith in the western continuation of the Gangdese belt[51]; and (3) the Nyingchi complex with coeval melting of crustal rocks at pressure of ~1.1 GPa between 69 and 41 Ma[54].

## Triggers for arc remelting

Although the change in the chemical composition of magmatism at ~70 Ma occurs before the timing of continental collision at 60–45 Ma, it coincides with: (1) a significant increase in the India-Asia convergence rate at 67 Ma lasting to 51 Ma[13] (Fig. 3), and (2) an intensification and southward migration of magmatism after the lull lasting between 80 and 70 Ma, which is ascribed to flat slab subduction[8,9,14]. These two changes have been linked to the rollback of the subducting Neo-Tethys oceanic slab[8,9,13,14]. Fast subduction is characterized by dehydration of the subducting slab, thereby increasing slab-derived fluid supply to the mantle wedge[41]. Slab rollback results in faster wedge corner flow velocities, hence increasing the temperatures of the mantle wedge[55]. The increased availability of hydrous fluids and higher temperatures in the mantle wedge substantially enhance the production of basaltic magmas by decompression melting[56], thereby triggering the partial melting of pre-existing arc lower crust. This mechanism readily explains the generation of K-rich felsic magmas with increased Th/La ratios between 67 and 51 Ma.

Subsequent rapid slowdown of the northward movement of the Indian plate at ~51 Ma (Fig. 3) has traditionally been interpreted as the onset of the India-Asia collision[57]. However, it is more likely caused by the loss of slab pull force, the main driving force of plate motion[58], due to slab breakoff[8,13]. Slab breakoff would occur at depths close to or shallower than the base of the overriding lithosphere[59] given that the Gangdese crust was > 50 km thick at that time[60,61]. Shallow breakoff results in the melting of the mantle wedge and the base of the overriding lithospheric mantle induced by the release of water from the tip of the detached slab as it heats up[59]. Such melting can produce high-K, high-$Al_2O_3$ calc-alkaline and/or shoshonitic basalts as represented by the ~51 Ma Upper Linzizong volcanic succession[28] and the coeval Gangdese mafic rocks (51–45 Ma). Injection of such basaltic magmas provide external heat as well as $H_2O$ that can depress the solidus temperature[29–31,46], resulting in extensive water-present remelting of the lower crust. Such basaltic magmatism will last for several million years, whereas the resulting crustal felsic magmatism related to remelting can proceed considerably longer[62], explaining the Gangdese magmatic flare-up at ~51 Ma and subsequent mafic and felsic magmatism (51–45 Ma).

The decrease in magma productivity accompanied by renewed increase in the values of $SiO_2$, $K_2O/SiO_2$, $K_2O/Na_2O$, and Th/La after ~45 Ma, matches the significant decrease in the convergence rate between India and Asia since ~45 Ma (Fig. 3). This deceleration is a result of the resistance of the Indian continental slab to subduction beneath the Gangdese belt[13]. Slow subduction increases the temperature of the subducting Indian slab for any given depth, allowing crustal melts being generated and transferred into the overriding mantle wedge and continental lithospheric mantle[41,63]. Decompression

melting of such metasomatized mantle due to lithospheric delamination[11] and/or tearing of the subducting Indian slab[64] could result in the generation of ultrapotassic magmas. These magmas could in turn provide the heat and water necessary for partial melting of the thickened Gangdese lower crust. Ultrapotassic and crustal felsic magmas could then mix, forming the post-collisional suite dominated by high-K calc-alkaline magmas.

## Density-sorting of the Gangdese continental crust

The fractional crystallization of pre-collisional arc magmas (Fig. 2a, b) resulted in the formation of abundant hornblende-rich cumulates (e.g., hornblendite and hornblende gabbro) (Supplementary Fig. 1). The long duration of the Gangdese arc magmatism suggests that these cumulates would be thicker than 30 km[45] by the late Cretaceous (Fig. 5a). It is commonly suggested for other continental arcs that cumulates may have been delaminated and recycled into the asthenospheric mantle[65]. If this was the case for the Gangdese belt, it should have occurred immediately before 70–68 Ma to account for the compositional changes. However, this is unlikely because (1) the ~80–70 Ma period was characterized by magmatic paucity with minor felsic magmatism[9,10,14], and (2) the 85–69 Ma period was marked by significant crustal shortening[9,14]. These are inconsistent with cumulate delamination models that predict the development of extensive magmatism and crustal extension[66]. Cumulate delamination during subsequent collision may be inhibited by partial subduction of incoming buoyant continental crust, providing a natural barrier for delamination[67].

The basaltic underplates and hornblende-rich cumulates in the Gangdese lower crust may have provided a fertile source[1,68] of felsic magmas when roll back of the subducting Neo-Tethyan oceanic slab started at ~70 Ma and then broke off at ~51 Ma[7–9,69]. Partial melting of the arc's lower crust to form felsic magmas is likely significant throughout continental arc magmatism[51,70,71]. However, it may gain in importance when significant thermal perturbations occur in response to changed mantle dynamics during late-subduction and collision[72,73]. Such remelting would further differentiate the continental arc crust by producing a felsic, potassic component that is transferred to the middle and upper crust (Fig. 5b), leaving refractory residues in the lower crust. The felsic component is preserved in the present-day Gangdese crust, where seismic data reveal the presence of a thick felsic crust[74]. The large volume of buoyant felsic melt removed from the lower crust after ~45 Ma would have increased the density of the refractory residue, but delamination may still have been prevented by the incoming buoyant Indian continental crust[67]. Instead, the densified lower crust would become part of the seismic sub-arc lithospheric mantle (Fig. 5b). This process is evidenced by the presence of ~85 Ma ultramafic xenoliths, consisting mainly of hornblende and biotite, that experienced granulite-facies metamorphism at 17–13 Ma, under P-T conditions of 19–24 kbar (~60–80 km) and ~1100 °C[75]. Such metamorphism and densification may now be recorded by a doublet Moho structure at depths of ~60–80 km[76] interpreted here to represent a refractory residual layer.

Thus, the magmatic activity in the Gangdese belt records an evolutionary path from a system dominated by fractionation and accumulation during oceanic subduction, to a system where remelting increased in importance, starting at ~70 Ma, during late-subduction and collisional times. Such accumulation-remelting sequence would be an efficient process of generating and strengthening chemical and density stratification, leading to a chemical maturation and thickening of juvenile continental crusts in collision zones, favoring their long-term preservation in the geological record.

## Implications for other continental collision zones

Hornblende-rich arc cumulates formed through fractional crystallization in the lower crust have been documented in both individual arcs[22,46,77] and global subduction zones[68,78]. These cumulates and underplated basaltic rocks provide a fertile source for remelting as a result of slab rollback and breakoff, which are inevitable during late-subduction and collision due to the density contrast between the subducting oceanic lithosphere and the attached buoyant continental block[59,79]. We also expect that collision zones in general will undergo a similar acceleration and deceleration in convergence rate immediately before and during collision, accompanied by thermal perturbations and water fluxing, as inferred here. This process could have taken place in many major collision zones throughout Earth's history, such as the East Kunlun[80], Eastern Pontides[81], European Variscides[82], Newfoundland Appalachians[83], and British Caledonides[84] (Fig. 1a). This is indicated by the presence of pre-collisional (ultra)mafic igneous rocks (including hornblende-rich cumulates) within granitic batholiths (Fig. 1a, b) and syn- and post-collisional felsic rocks, predominantly high-K to shoshonitic, similar to the Gangdese belt. The latter are interpreted as being derived from the remelting of old continental lower crust with minor addition of metasomatized lithospheric mantle material, geodynamically ascribed to slab rollback or breakoff[80,84–86].

## Data availability

All data supporting the findings of this study are provided in the Supplementary Materials.

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

## Acknowledgements

This paper is dedicated to China University of Geosciences, Beijing in celebration of its 70th birthday. This paper benefited from useful discussions with Sheng-Ao Liu, Jingao Liu, and Zeming Zhang. This research was financially supported by the National Natural Science Foundation of China Grants 91755207, 42121002, and 41225006, the Second Tibetan Plateau Scientific Expedition and Research Program (STEP) Grant 2019QZKK0702, the 111 project Grant B18048, and the Australian Research Council Grant FL160100168. This is China University of Geosciences (Beijing) petrogeochemical contribution PGC2015-098.

## Author contributions

D.C.Z. and Q.W. designed the project. D.C.Z., Q.W., and Z.Z. conducted fieldwork. Q.W. performed zircon U-Pb dating and zircon Hf-O isotopic determination. D.C.Z., Q.W., S.L.C., Z.Z., Z.Q.H. and X.X.M. wrote the first version of the manuscript. D.C.Z., Q.W., R.F.W., P.A.C., and Y.F.Z. contributed to data interpretation and writing the final version of the manuscript. All authors participated in the discussion and agreed on the content. We have considered local and regional research relevant to our study in the citations.

## Competing interests

The authors declare no competing interests.
