## [Peer Review File · Nature Communications]

REVIEWER COMMENTS

Reviewer #1 (Remarks to the Author):

The manuscript entitled "Building continental crust along convergent plate boundaries" is a very interesting paper regarding the continental crustal growth on our Earth. After compiling the available geochronological and geochemical data from the igneous rocks in the Gangdese, southern Tibet, it was proposed that the arc magmatism there can be divided into two stages of evolution, i.e., early fractionation of mantle-derived primitive arc rocks, and later re-melting of the early cumulate, which made the juvenile arc crust transformed into mature continental crust. The paper is well written, and the data are of high quality, hence I recommend its publication in our journal, but some revision is still needed.

Main comments:

1. Generally, the convergent plate boundary can be divided into the oceanic arc by oceanic subduction under another oceanic lithosphere, and continental arc by oceanic subduction under a continent. It was true that the Gangdese was mostly a continental arc (a convergent plate boundary) during the Mesozoic by subduction of the Tethyan ocean under the Asian continent, but it was also affected by collision of India to the south along the Yarlung Zangbo suture zone. Therefore, the Gangdese documented both oceanic subduction and later continental collision, and hence the title here is suggested to be modified to match the main contents of the text;
2. If the paper focused on the continental crustal evolution, it will be very constructive to compare the Gangdese with other arcs, especially with Cordillera and Andes along the eastern Pacific.

Specific comments:

1. Line 35-37: If the setting was initiated by early oceanic subduction and then followed by continental collision, this setting is better to ascribe as collisional, but not convergent boundary;
2. Line 50-53: For the Gangdese, an input of Indian continental material was undoubtedly involved during the Miocene magmatism, which cannot be ignored during the magmatic evolution of a continental collisional zone;
3. Line 59-61: Reference is needed;
4. Line 68: what is "accretionary orogen" here meant? An oceanic arc indeed?
5. Line 75: What criteria used here to define the "post-collisional";
6. Line 100 (and associated figure): the Z-shaped trend for the pre-collisional is not clearly identified if it was compared with the syn-, and post-collisional rocks;
7. Line 107-109: Please state the evidence that those pre-collisional mafic rocks are mantle-derived;
8. Line 138: this sub-section needs to be outlined, since there are several different kinds of rock assemblages discussed here;
9. Line 143-144: Is it possible that those ultramafic cumulates sank into mantle by delamination after its formation?
10. Line 145-149: If four types of rock assemblages can be classified, we should discuss their origination one by one;
11. Line 149-152: You mean that the gabbro is mantle-derived, but not a cumulate as well?
12. Line 162: what difference between "damp" and "wet", can you give the specific values here?
13. Line 164-165: Why the back-veining here is evidence of re-melting?
14. Line 169-171: If the post-collisional rocks were also from re-melting of pre-existing rocks, their Sr-Nd-Hf isotopic compositions will provide strong evidence, right?
15. Line 189-190: Please explain a little bit of "bivariate kernel density" here;
16. Line 216: From discussion of this sub-section, three stages of processes should be included, i.e., formation of the primitive arc, re-melting of the arc/cumulate (with in situ basement), and later input of subducted continent. The latter is the key difference between the studied Gangdese and the commonly assumed convergent plate boundaries;
17. Line 228-229: can we distinguish the subducted sediment with the pre-existed basement? Alternatively, is there any crustal (Asian or Indian) contamination during its

formation?

18. Line 290-292: it is better to have comparisons for the major and trace elements, but not only of zircon O isotopes;

19. Line 297: If no delamination processed, the whole crust will be still basaltic, which is not accord with the data presented, hence it is suggested to discuss a little bit this kind of process here;

20. Line 319-324: Can you provide the data of melting depths of these rocks?

21. Line 339-340: This kind of process is also documented in the Cordillera, does this mean that it is a common mechanism for continental maturation in chemistry?

22. Line 382-386: the evidences listed here are somewhat weak. Alternatively, the cumulate can sink to the mantle during the syn-collisional stage;

23. Line 391: More references are suggested for this conclusion;

24. Line 406-408: you mean a crust-mantle boundary?

25. Line 412-415: if this is correct, you have to apply delamination to keep the crust evolved toward more felsic;

26. Line 426: it is too concise to cite these collisional zones without detail descriptions;

27. Line 443-445: More explanations are necessary for a better understanding;

28. Line 445-447: as the key of the paper, it is better to specific the evolutionary processes according to their chronological order, such as the arc cumulation, slab deepening and re-melting of the lower base, continental input after collision, etc.

Fu-Yuan Wu

May 16, 2022

Reviewer #2 (Remarks to the Author):

Review for Zhu et al., Nature Communications

Zhu and his co-authors present a synthesis study on Gangdese magmatism and continental crust formation. The main idea of this paper is that melting of pre-existing cumulates during and after the continent collision phase is key to generating matured continental crust. Overall, this work is interesting and free of obvious pitfalls. My major criticism is that they talked too much, which makes this paper not easy to follow. The authors need to think about which observation provides the most important line of evidence instead of worrying about everything.

To me, the geochemical trends shown in Fig. 3 are particularly intriguing. I would suggest the authors work around these geochemical trends and show that these trends truly reflect changing sources from subduction to collision. For example, the authors may use quantitative geochemical modeling to test whether the increasing K/Si, K/Na, and Th/La in syn- and post-collision granites result from cumulate melting. MELTS simulations coupled with trace element calculations may be worth trying. Alternatively, the authors may take a look at existing experimental data and see if these data support their hypothesis.

Line 229. How do we know that the high d18O results from the addition of subducted sediments instead of assimilation in the arc crust?

Reviewer #3 (Remarks to the Author):

The paper is the result of examining a large number of samples from a geologically crucial place to understand the continental collision process and to illuminate the magmatism that accompanies the most important episodes of the Wilson cycle.

The dataset seems excellent using the proper methodology and excellent and up-to-date approaches. The synthesis of data is also very good and appropriate. However, the main problem I see with this paper is not the scientific contribution, but the way of presentation

and expression ie language (although I cannot fully judge the language). There is a number of problematic terms and sentences and I will only give a few of them:

-Line 217: "Arc magmas contain varying amounts of recycled pre-existing crustal material, including weathered oceanic or terrestrial sediments". I think should say that we talk about the recycled components derived from the crustal material.

-Line 225: here says that the "... the zircon $\delta^{18}O$ an effective monitor of the involvement of recycled pre-existing crustal materials with high and variable $\delta^{18}O$ due to low-temperature alteration" - it is not the alteration crustal lithologies the cause of the high $\delta^{18}O$, this is simplistic. Generally, the whole reasoning in lines 225-236 is not properly elaborated due to language problems, although I may agree with the author's ideas.

-line 272: the term "unfilled red diamonds"

-line 275: the term "unchanging zircon $\delta^{18}O$ " -should be invariable

-line 300: "If enough potassium has been introduced into such mantle-derived magma as it leaves the mantle wedge and the lithospheric mantle, it stalls at the base of the crust crystallizing to form a hornblende-rich cumulate (and then metamorphosed to amphibolite)". This is very a typical formulation that is illegible.

In summary, the science is excellent, the findings very relevant to the community, also the general structure of the paper. However, the manuscript desperately needs the language&expresion improvement, and therefore I suggest another round of review.

I hope that my review will be useful to the authors and editor and I wish them a good job.

Zhu et al.'s point-by-point response to the reviewers' comments

Revisions are marked with Yellow Background in the clear version and with tracked changes in a separate file.

Comments from Reviewer #1

General evaluation: The manuscript entitled “Building continental crust along convergent plate boundaries” is a very interesting paper regarding the continental crustal growth on our Earth. After compiling the available geochronological and geochemical data from the igneous rocks in the Gangdese, southern Tibet, it was proposed that the arc magmatism there can be divided into two stages of evolution, i.e., early fractionation of mantle-derived primitive arc rocks, and later re-melting of the early cumulate, which made the juvenile arc crust transformed into mature continental crust. The paper is well written, and the data are of high quality, hence I recommend its publication in our journal, but some revision is still needed.

Response: Thank you for this positive evaluation.

Main comments:

1. Generally, the convergent plate boundary can be divided into the oceanic arc by oceanic subduction under another oceanic lithosphere, and continental arc by oceanic subduction under a continent. It was true that the Gangdese was mostly a continental arc (a convergent plate boundary) during the Mesozoic by subduction of the Tethyan ocean under the Asian continent, but it was also affected by collision of India to the south along the Yarlung Zangbo suture zone. Therefore, the Gangdese documented both oceanic subduction and later continental collision, and hence the title here is suggested to be modified to match the main contents of the text.

Responses: Great comment. We have revised the title to “*Interplay between oceanic subduction and continental collision in building continental crust*”.

2. If the paper focused on the continental crustal evolution, it will be very constructive to compare the Gangdese with other arcs, especially with Cordillera and Andes along the eastern Pacific.

Responses: Our paper focuses on the crustal evolution from ocean subduction to continental collision. In addition to the comparisons with other similar tectonic setting (e.g., the East Kunlun magmatic belt in the northern Tibetan Plateau, the Odenwald in Variscan Europe of Germany, and the British Caledonides), we have also compared our work with the Chelan Complex of the incipient Cascades arc

(Dessimoz et al., 2012) and the mature magmatic arcs represented by the Sierra Nevada in North Cordillera (Lee et al., 2006). Please see lines 111-116 and 430-442.

Dessimoz, M., Müntener, O., and Ulmer, P., 2012, A case for hornblende dominated fractionation of arc magmas: The Chelan Complex (Washington Cascades): Contributions to Mineralogy and Petrology, v. 163, p. 567–589.

Lee, C. T. A., Cheng, X., and Horodyskyj, U., 2006, The development and refinement of continental arcs by primary basaltic magmatism, garnet pyroxenite accumulation, basaltic recharge and delamination: insights from the Sierra Nevada, California. Contributions to Mineralogy and Petrology 151, 222–242.

Specific comments:

1. Line 35-37: If the setting was initiated by early oceanic subduction and then followed by continental collision, this setting is better to ascribe as collisional, but not convergent boundary.

Response: we have revised “convergent boundary” to collision zones.

2. Line 50-53: For the Gangdese, an input of Indian continental material was undoubtedly involved during the Miocene magmatism, which cannot be ignored during the magmatic evolution of a continental collisional zone.

Response: We did not address this point in the abstract (i.e. line 46-52), but have discussed it in later sections (please see lines 191-196, 283-287, and 362-370). We have also added a subtitle to emphasize this point: “*Post-Collisional Suite Dominated by Remelting of Arc Crust Plus Subducted Indian Continental Material*” (Please see lines 181-182).

3. Line 59-61: Reference is needed.

Response: Thanks! Taylor & McLennan (1995) has been added.

Taylor, S. R., & McLennan, S. M. The geochemical evolution of the continental crust. *Rev. Geophys.* 40, 241-265 (1995).

4. Line 68: what is “accretionary orogen” here meant? An oceanic arc indeed?

Response: Accretionary orogen means oceanic subduction-related orogen (Cawood et al., 2009). Here it does not mean an oceanic arc, but a continental arc as stated in line 68. For clarifying this point, we have: (1) re-worded this sentence (Please see lines 66-69) and (2) added one word “*continental*” in

the abstract (Please see line 41).

Cawood, P. A., Kröner, A., Collins, W. J., Kusky, T. M., Mooney, W. D., & Windley, B. F. Accretionary orogens through Earth history. Geological Society, London, Special Publications, 318(1), 1–36 (2009).

5. Line 75: What criteria used here to define the “post-collisional”.

Response: According to Zhu et al. (2015, Scientific Reports), “post-collisional” is defined as the time following complete loss of the pull force from the subducting oceanic slab due to slab breakoff. We have re-phased the sentence that now reads: “*This subdivision is based on the multiple lines of evidence that constrain the timing of initial collision between India and Asia to ~60 Ma^{15,16}. It is also based on post-collision being defined as the time following complete loss of the pull force from the subducting oceanic slab due to its breakoff, which started at ~45 Ma⁸⁷*”. Please see lines 78-81.

Zhu, D. C., Wang, Q., Zhao, Z. D., Chung, S. L., Cawood, P. A., Niu, Y. L., Liu, S. A., Wu, F. Y. & Mo, X. X. Magmatic record of India-Asia collision. *Sci. Rep.* **5**, 14289 (2015).

6. Line 100 (and associated figure): the Z-shaped trend for the pre-collisional is not clearly identified if it was compared with the syn-, and post-collisional rocks.

Response: The Z-shaped trend is defined by the presence of dunite to wehrlite, pyroxenite, hornblendite samples, which are not documented by the syn-, and post-collisional rocks. This trend may be influenced by the limited number of dunite to wehrlite, pyroxenite samples. This has been clarified in lines 123-127.

7. Line 107-109: Please state the evidence that those pre-collisional mafic rocks are mantle-derived.

Response: We have added the following two references to indicate that those pre-collisional mafic rocks are mantle-derived. Please see line 117.

Xu, W., Zhu, D. C., Wang, Q., Weinberg, R. F., Wang, R., Li, S. M., Zhang, L. L. & Zhao, Z. D. Constructing the Early Mesozoic Gangdese crust in southern Tibet by hornblende-dominated magmatic differentiation. *J. Petrol.* **60**, 515–55 (2019).

Guo, L., Jagoutz, O., Shinevar, W. J. & Zhang, H. F. Formation and composition of the Late Cretaceous Gangdese arc lower crust in southern Tibet. *Contrib. Mineral. Petrol.* **175**, 58 (2020).

8. Line 138: this sub-section needs to be outlined, since there are several different kinds of rock assemblages discussed here.

Response: We have split this sub-section into two sub-sections: “*Syn-Collisional Suite Dominated by Remelting of Arc Crust*” and “*Post-Collisional Suite Dominated by Remelting of Arc Crust Plus Subducted Indian Continental Material*”. Please see line 147 and lines 181-182.

9. Line 143-144: Is it possible that those ultramafic cumulates sank into mantle by delamination after its formation?

Response: We think this is less likely. We have discussed this possibility in a later section (lines 377-386).

10. Line 145-149: If four types of rock assemblages can be classified, we should discuss their origination one by one.

Response: We have provided simple explanations for their origins as follows: (1) small amounts of hornblendites and cumulate hornblende gabbros *suggesting limited accumulation*, (2) widespread 58–45 Ma mafic microgranular enclaves and dykes *indicative of basaltic injections and magma mixing/mingling*^{8–10}, (3) voluminous high-K felsic rocks ($\text{SiO}_2 > 60 \text{ wt}\%$) (Fig. 2f). Please see lines 153-156.

11. Line 149-152: You mean that the gabbro is mantle-derived, but not a cumulate as well?

Response: Gabbro is mantle derived and either cumulate or non-cumulate. We have added the word “*cumulate*” to indicate the cumulitic nature of the hornblende gabbro mentioned here (in line 157).

12. Line 162: what difference between “damp” and “wet”, can you give the specific values here?

Response: We have provided the specific values and related references in line 172-173.

13. Line 164-165: Why the back-veining here is evidence of re-melting?

Response: The dioritic dyke is younger and hotter than the wall rock (tonalite), Its emplacement resulted in the re-melting of the wall rock to form back-veining. We have re-organized this sentence to clarify this point: “*Field*

evidence for melting is shown by back-veining of a ~48 Ma dioritic dyke by felsic magma derived from the melting of the ~79 Ma wall-rock". Please see lines 175-176.

14. Line 169-171: If the post-collisional rocks were also from re-melting of pre-existing rocks, their Sr-Nd-Hf isotopic compositions will provide strong evidence, right?

Response: Yes, this is precisely the case for felsic rocks older than 30 Ma as indicated by the similar zircon $\delta^{18}\text{O}$, but not the case for felsic rocks younger than 30 Ma that contain increased amounts of the subducted Indian slab-derived materials. These points have been addressed in later sections (lines 191-196 and 283-287).

15. Line 189-190: Please explain a little bit of “bivariate kernel density” here.

Response: Kernel density estimation is a nonparametric technique for density estimation. It is a popular tool for visualising the distribution of data. We have added the explanation: “*Colored background indicates sample density distribution as measured by bivariate kernel density, where redder background corresponds to increased sample concentration or density*” in the caption of **Figure 3** to explain the “bivariate kernel density”. Please see line 221-223.

16. Line 216: From discussion of this sub-section, three stages of processes should be included, i.e., formation of the primitive arc, re-melting of the arc/cumulate (with in situ basement), and later input of subducted continent. The latter is the key difference between the studied Gangdese and the commonly assumed convergent plate boundaries.

Response: We have re-written this section using three paragraphs to emphasize the influences of Recycled Supracrustal Component for pre-collisional, syn-collisional, and for post-collisional samples. Please see lines 235-248, 272-282, and 283-287.

17. Line 228-229: can we distinguish the subducted sediment with the pre-existed basement? Alternatively, is there any crustal (Asian or Indian) contamination during its formation?

Response: (1) Distinguishing the subducted sediment from the pre-existing overriding basement has been a significant challenge. However, it remains possible by

mixing modelling of different end members that have been carefully examined (DePaolo & Wasserburg, 1979; Elburg & Foden, 1999; Jagoutz et al., 2019). (2) The curvature of mixing curves and thus the amounts of each end member are controlled by the relative abundances of trace elements of the selected end members. In this case, to distinguish the subducted sediment from the pre-existing Gangdese basement, we use the Hf abundances and Hf-O isotopic compositions of each local end member as provided in **Supplementary Materials**. (3) The methods for selecting the Hf abundances and Hf-O isotopic compositions of each end-member are given in **Supplementary Materials**.

DePaolo, D.J., Wasserburg, G.J., 1979. Petrogenetic mixing models and Nd-Sr isotopic patterns. *Geochim. Cosmochim. Acta* **43**, 615–627.

Elburg MA, Foden J. 1999. Geochemical response to varying tectonic settings: an example from southern Sulawesi (Indonesia). *Geochim. Cosmochim. Acta* **63**, 1155–1172.

Jagoutz O, Bouilhol P, Schaltegger U, Müntener O. 2019. The isotopic evolution of the Kohistan Ladakh arc from subduction initiation to continent arc collision. *Spec. Publ. - Geol. Soc. London*. **483**, 165–182.

18. Line 290-292: it is better to have comparisons for the major and trace elements, but not only of zircon O isotopes.

Response: We have carefully checked the paper we cited here (Lackey et al., 2005) and references therein to find out major and trace element data of samples that have been analyzed for zircon O isotope by Lackey et al. (2005). However, the authors did not report the major and trace element data but just introduced the lithology. This makes them unavailable to compare with the Gangdese samples. To respond to this comment, therefore, we have added the lithologies (i.e. gabbros and granitoids from the Sierra Nevada batholith) to the main text. Please see lines 290.

Lackey, J. S., Valley, J. W. & Saleeby, J. B. Supracrustal input to magmas in the deep crust of Sierra Nevada batholith: evidence from high- $\delta^{18}\text{O}$ zircon. *Earth Planet. Sci. Lett.* **235**, 315–330 (2005).

19. Line 297: If no delamination processed, the whole crust will be still basaltic, which is not accord with the data presented, hence it is suggested to discuss a little bit this kind of process here.

Response: We discussed delamination later in lines 377-386 and 393-407. To avoid repetition, we did not discuss it here.

20. Line 319-324: Can you provide the data of melting depths of these rocks?

Response: We have added pressure data available for these rocks. Please see line 320 and line 324-325.

21. Line 339-340: This kind of process is also documented in the Cordillera, does this mean that it is a common mechanism for continental maturation in chemistry?

Response: Although this mechanism is not so common, it has been proposed to explain the magmatic flare-up at 103–94 Ma in the early Andean Cordillera, which is caused by increasing contributions of slab-derived fluids and decreasing volumes of sediments as a result of high convergence rates and relative decoupling of plates (Jara et al., 2021). This mechanism is also consistent with previous studies on modern volcanic arcs, which reveal a positive correlation between caldera occurrence and convergence rate (except in arcs with back-arc spreading) (Hughes and Mahood, 2008).

To respond to this comment, we have added “*We also expect that collision zones in general will undergo a similar acceleration and deceleration in convergence rate immediately before and during collision, accompanied by thermal perturbations and water fluxing, as recorded here*” to lines 430-433.

Hughes GR, Mahood GA. 2008. Tectonic controls on the nature of large silicic calderas in volcanic arcs. *Geology* 36: 627–630.

Chapman JB, Shields JE, Ducea MN, Paterson SR, Attia S, Ardill KE. 2021. The causes of continental arc flare ups and drivers of episodic magmatic activity in Cordilleran orogenic systems. *Lithos* 398–399: 106307.

Jara JJ, Barra F, Reich M, Leisen M, Romero R, Morata D. 2021. Episodic construction of the early Andean Cordillera unravelled by zircon petrochronology. *Nat. Commun.* 12: 4930.

22. Line 382-386: the evidences listed here are somewhat weak. Alternatively, the cumulate can sink to the mantle during the syn-collisional stage.

Response: It is difficult to verify whether the cumulate has been delaminated or not in ancient orogenic belts. To enhance our argument, we added an additional point to lines 384-386: “*Cumulate delamination during subsequent collision may be inhibited by partial subduction of incoming buoyant continental crust, providing a natural barrier for delamination*⁶⁸”.

Ganade, C. E., Lanari, P., Rubatto, D., Herman, J., Weinberg, R. F., Basei, M. A. S., Tesser, L. R., Caby, R., Agbossoumondé, Y. & Ribeiro, C. M. Magmatic flare-up causes crustal thickening at the transition from subduction to continental collision. *Commun. Earth Environ.* 2, 41 (2021).

23. Line 391: More references are suggested for this conclusion.

Response: Two references have been added: one focused on the anatexis of the Gangdese arc during the Late Cretaceous (Ding et al., 2022, JPet) and the other on the remelting of earlier intrusive rocks driven by fluctuations in pressure or temperature from the Ladakh Batholith in NW Himalayas (Weinberg and Dunlap, 2000). Please see line 391.

Ding, H. X., Zhang, Z. M., Palin, R. M., Kohn, M. J., Niu, Z., et al. 2022. Late Cretaceous metamorphism and anatexis of the Gangdese magmatic arc, south Tibet: Implications for thickening and differentiation of juvenile crust. *J. Petrol.* **63**: egac017.

Weinberg, R. F. and Dunlap, W. J. Growth and deformation of the Ladakh Batholith, Northwest Himalayas: Implications for timing of continental collision and origin of calc-alkaline batholiths. *The Journal of Geology* 108 (3), 303-320.

24. Line 406-408: you mean a crust-mantle boundary?

Response: Yes, the lower and upper boundaries of such doublet Moho structure are suggested to indicate the petrological and seismic Moho, respectively. To clarify this point, we added “seismic” to line 401.

25. Line 412-415: if this is correct, you have to apply delamination to keep the crust evolved toward more felsic.

Response: We feel that there is no need to delaminate the refractory ultramafic-mafic residual layer because, as mentioned, the incoming buoyant continental crust could provide a natural barrier for delamination (Ganade et al., 2021). Instead, such residual layer may have seismically become part of the sub-arc lithospheric mantle, resulting in the vertical chemical and density stratification of continental crust in collision zones. The Gangdese crust could be the best example of such interpretation.

Ganade, C. E., Lanari, P., Rubatto, D., Herman, J., Weinberg, R. F., Basei, M. A. S., Tesser, L. R., Caby, R., Agbossoumondé, Y. & Ribeiro, C. M. Magmatic flare-up causes crustal thickening at the transition from subduction to continental collision. *Commun. Earth Environ.* **2**, 41 (2021).

26. Line 426: it is too concise to cite these collisional zones without detail descriptions.

Response: (1) The difficulty we find in providing more details is that cumulates within granitic batholiths are often not included in the dataset and their genetic relations to granitic batholiths are typically unknown. (2) A more detailed

description of these collision zones is introduced in a companion paper (Zhu et al., Under Review).

Zhu, D.C., Wang, Q., Weinberg, R. F., Cawood, P.A., Zhao, Z.D., Hou, Z.Q., Mo, X.X., Continental crustal growth processes recorded in the Gangdese Batholith, southern Tibet. *Annu. Rev. Earth Planet. Sci.*, Under Review.

27. Line 443-445: More explanations are necessary for a better understanding.

Response: Given the limited data available for ultramafic and mafic rocks in the literature, we have added a few more explanations in terms of external forces (e.g., convergence acceleration and deceleration) and internal factors (e.g., thermal perturbations and water fluxing). Please see lines 430-433.

28. Line 445-447: as the key of the paper, it is better to specific the evolutionary processes according to their chronological order, such as the arc cumulation, slab deepening and re-melting of the lower base, continental input after collision, etc.

Response: We have re-organized this subsection along chronological order. Please see lines 425-442.

Comments from Reviewer #2

Zhu and his co-authors present a synthesis study on Gangdese magmatism and continental crust formation. The main idea of this paper is that melting of pre-existing cumulates during and after the continent collision phase is key to generating matured continental crust. Overall, this work is interesting and free of obvious pitfalls. *My major criticism is that they talked too much, which makes this paper not easy to follow. The authors need to think about which observation provides the most important line of evidence instead of worrying about everything.*

Response: Many Thanks! We have moved the details related to binary mixing modelling and the introduction to the rationale to the Supplementary Materials and re-organized the section of “Role of Recycled Supracrustal Component in Generating the Compositional Change”. It has been reduced from 998 to 768 words.

To me, the geochemical trends shown in Fig. 3 are particularly intriguing. I would suggest the authors work around these geochemical trends and show that these trends

truly reflect changing sources from subduction to collision. For example, the authors may use quantitative geochemical modeling to test whether the increasing K/Si, K/Na, and Th/La in syn- and post-collision granites result from cumulate melting. *(1) MELTS simulations coupled with trace element calculations may be worth trying. Alternatively, (2) the authors may take a look at existing experimental data and see if these data support their hypothesis.*

Response: Great comments! **Regarding the first comment:**

To test whether the increase in K/Si, K/Na, and Th/La in syn- and post-collision granites result from cumulate melting, we used a hornblende gabbro with 49.71 wt% SiO₂ and 2.28 wt% K₂O as the starting material to simulate the melt composition during partial melting using MELTS. The reasons for selecting this high-K sample include: (1) basaltic underplates and cumulates from global continental arcs (including the Gangdese continental arc) are characterized by high K₂O contents (mostly > 0.5 wt.%) compared to global oceanic arcs, and (2) experimental studies (Dufek & Bergantz, 2005; Sisson et al., 2005) and thermodynamic modeling (Wang et al., 2022) show that the K₂O contents of the partial melts are directly linked with those of the starting materials, regardless of cumulate and noncumulate origin.

We set up the conditions of simulation at 2.0 GPa and 800–1000 °C adding 2 wt% H₂O, which are within the range of P-T conditions expected for the lowermost arc crust (~800–1000 °C at 35–70 km depth; Ducea et al., 2021).

The results in the figures below show that in the temperature range of ~800–1000 °C, corresponding to melt fractions of 20–30%, the produced melts have high [K₂O/SiO₂]_c ratios >4 and high [Th/La]_c ratios >2 (Figure 1 below), higher than the median [K₂O/SiO₂]_c shown in Figure 2b or close to the median [Th/La]_c in Figure 3d in the main text. Therefore, the increased K₂O/SiO₂ ratios since 70 Ma could be the combined result of biotite ± hornblende breakdown as they are the main K-bearing mineral phases in basaltic underplates and cumulates, whereas the increased Th/La ratios are most likely controlled by the difference of incompatibility between Th and La, i.e., Th is more incompatible than La and thus more preferentially concentrates in the melt during melting.

We have revised the main text (Please see lines 303 and 311-317), citing the results of thermodynamic and trace element modeling (Wang et al., 2022).

Figure 1 The compositions produced by MELTS at 2.0 GPa and 800-1000 °C adding 2 wt% H₂O for a hornblende gabbro (49.71 wt% SiO₂ and 2.28 wt% K₂O). It shows that the produced melts have high $[K_2O/SiO_2]_c$ ratios > 4 and high $[Th/La]_c$ ratios > 2 in the temperature range of ~800–1000 °C that corresponds to 20–30% degree of melting. The grey field corresponds to temperature range of the lowermost arc crust at ~800–1000 °C and 35–70 km depth (Ducea et al., 2021).

Pertaining the second comment: (1) Dehydration melting experiments have indicated that K₂O contents of partial melts are directly linked with those of the starting materials, regardless of whether they have a cumulate (Dufek & Bergantz, 2005) or noncumulate origin (Sisson et al., 2005). (2) Our unpublished data from dehydration melting experiments show that natural hornblende-rich cumulate samples, common in arc lower crusts, can produce shoshonitic to medium-K felsic melt compositions at 900–1000 °C and 1.0–2.0 GPa (Figure 2). Existing experimental data are shown for comparison in Figure 2. In the revision, however, (1) we did not present the results of our unpublished experimental data and the existing experimental data, to avoid complexity, and (2) we have re-organized this subsection by highlighting experimental studies and thermodynamic modeling (Wang et al., 2022). Please see line 303.

Figure 2 Plots of CIPW normative quartz, orthoclase, and albite + anorthite (A) and of K₂O versus SiO₂ (B). Regardless of temperature and pressure, the low-K₂O sample (17ML01-1) produced medium-K monzogranitic to granodioritic melts, and the two high-K₂O samples (16CJ17-7 and 16CJ17-9) produced shoshonitic to high-K monzogranitic, granodioritic, and monzodioritic melts (A-B). Results of previous experiments on K-rich (Sen and Dunn, 1994; Sisson et al., 2005; Xiong et al., 2005) and K-poor (Wolf and Wyllie, 1994; Blatter et al., 2013; Qian and Hermann, 2013) starting materials are shown in A-B for comparison.

Blatter, D., Sisson, T., Hankins, W.B., 2013, Crystallization of oxidized, moderately hydrous arc basalt at mid- to lower-crustal pressures: Implications for andesite genesis: Contributions to Mineralogy and Petrology, v. 166, p. 861–886.

Ducea, M. N., Chapman, A. D., Bowman, E., and Triantafyllou, A., 2021, Arclogites and their role in continental evolution; part 1: Background, locations, petrography, geochemistry, chronology and thermobarometry: Earth-Science Reviews, v. 214, p. 1–11.

Dufek J, Bergantz GW. 2005. Lower crustal magma genesis and preservation: a stochastic framework for the evaluation of basalt-crust interaction. *J. Petrol.* 46:2167–95

Qian, Q., and Hermann J., 2013, Partial melting of lower crust at 10–15 kbar: Constraints on adakite and TTG formation: Contributions to Mineralogy and Petrology, v. 165, p. 1195–1224.

Sen, C., and Dunn, T., 1994, Dehydration melting of a basaltic composition amphibolite at 1.5 and 2.0 GPa: Implications for the origin of adakites: Contributions to Mineralogy and Petrology, v. 117, p. 394–409.

Sisson, T.W., Ratajeski, K., Hankins, W.B., and Glazner, A.F., 2005, Voluminous granitic magmas from common basaltic sources: Contributions to Mineralogy and Petrology, v. 148, p. 635–661.

Wang, X. S., Sun, M., Weinberg, R. F., Cai, K. D., Zhao, G. C., Xia, X. P., Li, P. F., Liu, X. J. Adakite

- generation as a result of fluid-fluxed melting at normal lower crustal pressures. *Earth and Planetary Science Letters* 594, 117744 (2022).
- Wolfe, M.B., and Wyllie, P.J., 1994, Dehydration-melting of amphibolite at 10 kbar: the effects of temperature and time: *Contributions to Mineralogy and Petrology*, v. 115, p. 369–383.
- Xiong, X.L., Adam, T.J., and Green, T.H., 2005, Rutile stability and rutile/melt HFSE partitioning during partial melting of hydrous basalt: Implications for TTG genesis: *Chemical Geology*, v. 218, p. 339–359.

Line 229. How do we know that the high $d^{18}\text{O}$ results from the addition of subducted sediments instead of assimilation in the arc crust?

Response: This comment is similar to the Comment 17 by the first reviewer. Distinguishing the addition of subducted sediment from the pre-existing overriding basement has been a significant challenge. However, it remains possible to use mixing modelling of different end members as has been carefully examined (DePaolo & Wasserburg, 1979; Elburg & Foden, 1999; Jagoutz et al., 2019). This is because the curvature of the mixing curves, and thus the amounts of each end member, is controlled by the relative abundances of trace elements of the selected end members. We have addressed these points in the Supplementary Materials.

Comments from Reviewer #3

The paper is the result of examining a large number of samples from a geologically crucial place to understand the continental collision process and to illuminate the magmatism that accompanies the most important episodes of the Wilson cycle.

The dataset seems excellent using the proper methodology and excellent and up-to-date approaches. The synthesis of data is also very good and appropriate. However, the main problem I see with this paper is not the scientific contribution, but the way of presentation and expression i.e. language (although I cannot fully judge the language). There is a number of problematic terms and sentences, and I will only give a few of them:

-Line 217: "*Arc magmas contain varying amounts of recycled pre-existing crustal material, including weathered oceanic or terrestrial sediments*". I think should say that we talk about the recycled components derived from the crustal material.

Response: We have re-written this paragraph. Please see lines 235-248.

-Line 225: here says that the "... the zircon $\delta^{18}\text{O}$ an effective monitor of the involvement of recycled pre-existing crustal materials with high and variable $\delta^{18}\text{O}$ due to low-temperature alteration" - it is not the alteration crustal lithologies the cause of the high $\delta^{18}\text{O}$, this is simplistic. Generally, the whole reasoning in lines 225-236 is not properly elaborated due to language problems, although I may agree with the author's ideas.

Response: To make it clearer, we have moved this sentence to the Supplementary Material: "*However, the exact amounts of sediment contribution are difficult to constrain because of the variable $\delta^{18}\text{O}$ values of subducted oceanic sediments varying from 10 to 40%⁴⁰*".

-line 272: the term "unfilled red diamonds"

Response: This term has been revised to "*red diamonds*". Please see line 247.

-line 275: the term "unchanging zircon $\delta^{18}\text{O}$ " - should be invariable

Response: It has been revised to "*the nearly unchanged zircon $\delta^{18}\text{O}$* ". Please see line 272.

-line 300: "If enough potassium has been introduced into such mantle-derived magma as it leaves the mantle wedge and the lithospheric mantle, it stalls at the base of the crust crystallizing to form a hornblende-rich cumulate (and then metamorphosed to amphibolite)". This is very a typical formulation that is illegible.

Response: Yes, the sentence was unclear. We have re-phased this sentence to "*If enough potassium has been incorporated into such a mantle-derived magma due to melt metasomatism from subducted sediment, it may form a hornblende-rich cumulate (metamorphosed to amphibolite) when it stalls at the base of the crust⁴⁸*". Please see lines 300-303.

In summary, the science is excellent, the findings very relevant to the community, also the general structure of the paper. However, the manuscript desperately needs the language & expression improvement, and therefore I suggest another round of review.

Response: Many Thanks for this positive evaluation! We have tried our best to improve the language & expression, including several rounds of careful edits by Roberto Weinberg and Peter Cawood. We hope that you will find the

manuscript has been improved significantly.

I hope that my review will be useful to the authors and editor and I wish them a good job.

Response: Your review is very helpful and useful for our revision. Thank you so much!

REVIEWERS' COMMENTS

Reviewer #1 (Remarks to the Author):

I read the revised manuscript, and was content with the modifications made, hence I recommend its publication in our journal.

Reviewer #2 (Remarks to the Author):

I'm satisfied with the revisions made by the authors and have no further comments.

Reviewer #3 (Remarks to the Author):

This is the revised version of the manuscript I already reviewed.

The paper is significantly improved, and now it is much more readable. However, there are still several inconsistencies:

1. The explanation about the kernel density presentation should already come in the caption of Fig.2.
2. line 286-please remove one "involvement".
3. throughout the entire manuscript, partial explanations exist about the character of the mantle source from which the basaltoid magmas of all three groups of rocks originate. However, I think it should be clearly written in the Results section, e.g. in the sections dedicated to each individual group of rocks.
4. Authors should avoid sentences that begin by negating some ideas and then explain which ideas they support in the second part; I think it should be just the opposite! A typical example is line 273: "This coincides not with the involvement of the Gangdese ancient basement materials but with 0–10 wt% input of Indian crust"

Sincerely yours
Prelević Dejan

Zhu et al.'s point-by-point response to the reviewers' comments

Revisions are marked with Yellow Background in the clear version.

Reviewer #1 (Remarks to the Author):

I read the revised manuscript, and was content with the modifications made, hence I recommend its publication in our journal.

Response: Thank you for this positive evaluation.

Reviewer #2 (Remarks to the Author):

I'm satisfied with the revisions made by the authors and have no further comments.

Response: Thank you for this positive evaluation.

Reviewer #3 (Remarks to the Author)

This is the revised version of the manuscript I already reviewed.

The paper is significantly improved, and now it is much more readable. However, there are still several inconsistencies:

1. The explanation about the kernel density presentation should already come in the caption of Fig. 2.

Responses: The explanation of “Colored background indicates sample density distribution as measured by bivariate kernel density, where red background corresponds to increased sample concentration or density” has been moved to the caption of Fig. 2. Please see lines 128-130.

We also added “The same is true for Fig. 3” to line 130 to explain the kernel density in Fig. 3.

2. line 286-please remove one "involvement".

Responses: Done.

3. throughout the entire manuscript, partial explanations exist about the character of the mantle source from which the basaltoid magmas of all three groups of rocks

originate. However, I think it should be clearly written in the Results section, e.g. in the sections dedicated to each individual group of rocks.

Responses: We have carefully checked the results section and references to basalts derived from the mantle. We feel that the current text explains how magmatism changed from wet basalts in pre-collisional times, followed by damp basalts during collision typical of arc magmatism, and then by ultrapotassic magmas derived from a lithospheric Indian mantle after collision.

To further clarify this point, we added words “wet” and “basaltic melts from the metasomatized mantle wedge” to lines 111-112. Please compare the following:

Previous: *These results indicate that the pre-collisional mafic rocks were derived from fractional crystallization of mantle-derived melts^{17,18} with high initial H₂O content, likely in excess of 3.0 wt%^{24,25} in an immature arc.*

Now: *These results indicate that the pre-collisional mafic rocks were derived from wet fractional crystallization of basaltic melts from the metasomatized mantle wedge^{17,18} with high initial H₂O content, likely in excess of 3.0 wt%^{24,25} in an immature arc.*

4. Authors should avoid sentences that begin by negating some ideas and then explain which ideas they support in the second part; I think it should be just the opposite! A typical example is line 273: "This coincides not with the involvement of the Gangdese ancient basement materials but with 0–10 wt% input of Indian crust"

Responses: We have revised this sentence to “*This coincides with 0–10 wt% input of Indian crust, rather than the involvement of the Gangdese ancient basement materials (compare Curves C and D in Fig. 4d)*”. Please see lines 265-266.

Similarly, we have also revised “*The change in the chemical composition of magmatism at 70–68 Ma coincides, not with the timing of continental collision at 60–45 Ma, but with:*” in lines 320-321 to “*Although the change in the chemical composition of magmatism at 70–68 Ma occurs before the timing of continental collision at 60–45 Ma, it coincides with:*”.